# Sensory Nutrition and Bitterness and Astringency of Polyphenols

**DOI:** 10.3390/biom14020234

**Published:** 2024-02-17

**Authors:** Naomi Osakabe, Takafumi Shimizu, Yasuyuki Fujii, Taiki Fushimi, Vittorio Calabrese

**Affiliations:** 1Functional Control Systems, Graduate School of Engineering and Science, Shibaura Institute of Technology, Tokyo 135-8548, Japan; 2Systems Engineering and Science, Graduate School of Engineering and Science, Shibaura Institute of Technology, Tokyo 135-8548, Japan; nb21110@shibaura-it.ac.jp; 3Department of Bio-Science and Engineering, Faculty of System Science and Engineering, Shibaura Institute of Technology, Tokyo 135-8548, Japan; bn20003@shibaura-it.ac.jp (T.S.); fujii.yasuyuki.x1@shibaura-it.ac.jp (Y.F.); 4Department of Biomedical and Biotechnological Sciences, University of Catania, 95125 Catania, Italy; vittorio.calabrese@unict.it

**Keywords:** polyphenol, sensory nutrition, bitterness, astringency, central nervous system

## Abstract

Recent studies have demonstrated that the interaction of dietary constituents with taste and olfactory receptors and nociceptors expressed in the oral cavity, nasal cavity and gastrointestinal tract regulate homeostasis through activation of the neuroendocrine system. Polyphenols, of which 8000 have been identified to date, represent the greatest diversity of secondary metabolites in plants, most of which are bitter and some of them astringent. Epidemiological studies have shown that polyphenol intake contributes to maintaining and improving cardiovascular, cognitive and sensory health. However, because polyphenols have very low bioavailability, the mechanisms of their beneficial effects are unknown. In this review, we focused on the taste of polyphenols from the perspective of sensory nutrition, summarized the results of previous studies on their relationship with bioregulation and discussed their future potential.

## 1. Introduction

Polyphenols are the largest and most diverse group of plant secondary metabolites, with 8000 identified to date [1]. Polyphenols are known to be coloring [2], bitter [3,4] and astringent [5,6] substances and have a significant impact on food palatability. In addition, daily consumption of these beverages can transform astringency and bitterness, which are inherently aversive perceptions, into pleasurable stimuli [7,8,9].These alterations suggest that changes in taste preference and avoidance due to long-term taste exposure can be considered as adaptations accompanied by neuroplasticity. However, the mechanisms by which taste preferences change over time are not yet understood. 

Epidemiological studies have reported a negative correlation between polyphenol intake and cardiovascular disease [10], neurodegenerative diseases [11,12] and age-related deterioration of sensory organs [13]. There are reports that people who are regular consumers of astringent and bitter drinks may be less at risk of type 2 diabetes and cardiovascular diseases, such as coffee and tea [14,15]. Large-scale intake studies of polyphenols from cocoa have also shown reduced cardiovascular deaths [16] and hippocampus-dependent cognitive improvements in the elderly [17]. 

In general, polyphenols are rarely absorbed from the upper gastrointestinal tract and move to the lower gut, where they are partly broken down by intestinal bacteria, but mostly excreted in the feces [18]. Their beneficial mechanism of action is not known because it is very unlikely to be distributed in the blood or organs, including the brain. Recent studies have reported that polyphenols ingested from the diet can alter the composition of the gut flora [19]. It has been reported that the composition of secondary metabolites in the colon changes accordingly and that these may be absorbed and affect cardiovascular disease and cognitive function [20]. On the other hand, it is extremely difficult to elucidate these causal relationships because the type and quantity of polyphenols ingested vary greatly with diet, and the quality and quantity of polyphenol metabolites produced by the gut microbiota and themselves vary widely between individuals.

This review, therefore, focuses on the interaction of bitter and astringent perceptions of polyphenols with sensory receptors, particularly with those that are expressed in the digestive tract other than the oral cavity. In addition, we summarized the results of previous studies on how signals produced by polyphenols affect the central nervous systems (CNS) via gastrointestinal sensory receptors and how such signals affect peripheral organs and explored the mechanisms involved.

## 2. Receptors Involved in Sensory Nutrition

In recent years, attention has focused on sensory nutrition, a field of study that examines how the senses conveyed by what people eat and drink act on the brain and how these signals affect human behavior and homeostasis [21]. In 2019, the National Institutes of Health (NIH) hosted a workshop on “Sensory Nutrition and Disease”. The workshop invited a diverse group of researchers from neuroscience, food science, psychology, nutrition and health sciences to understand how chemosensory influences affect eating. The workshop encouraged people to explore how these influences impact their choices and health [22]. Three topics were discussed at the conference: (a) the need to optimize chemosensory testing and assessment in humans; (b) the plasticity of the chemosensory system; and (c) the interplay between chemosensory signals, cognitive signals, dietary intake and metabolism, providing some guidance in advancing sensory nutrition research.

Chemosensory receptors expressed in the oral-nasal trigeminal system have been well studied to transmit signals from food to the brain and determine feeding behavior, such as ingestion or rejection [23,24]. Olfactory receptors respond to thousands of different types of volatiles [25,26,27]. Taste receptors respond to salts, sugars or sweeteners, amino acids, alkaloids, various bitter chemicals such as cycloheximide propylthiouracil denatonium benzoate, strychnine, etc., acids and fats. In addition, stimuli such as temperature, osmotic pressure, pH, pungency and many volatile compounds such as menthol aldehyde relatives, anesthetics, nicotine, etc., are recognized by transient receptor potential (TRP) channels [28,29,30,31]. Flavor perception is formed by integrating signals from various chemosensory systems in the oral and nasal cavities within the brain [32,33]. Flavor perception is thought to be shaped by neural processes occurring in chemosensory regions of the brain, including the anterior insula, frontal operculum, orbitofrontal cortex and anterior cingulate cortex and by interactions with other heteromodal areas including the posterior parietal cortex and ventral lateral prefrontal cortex. In recent years, it has been discovered that chemoreceptors are expressed outside of the oral-nasal trigeminal system [24,34,35]. For instance, it is known that olfactory receptors are in sperm regulating their motility germ cells [36], in skeletal muscle controlled differentiation and regeneration [37], in adipocytes [38] and liver [39] regulated energy metabolism and involved in gastrointestinal hormone secretion in the intestinal tract [40]. The sweet taste receptor 2/3 (T1R2/3) is expressed in various parts of the body, including the gut, pancreas, brain, bladder, bone and adipose tissue and plays a role in energy metabolism [41]. The bitter taste receptor, the taste receptor 2 (T2R) receptors, is expressed in several tissues, including the lungs, gastrointestinal tract, kidneys, genitals and brain. In addition, it has been shown to induce strong relaxation in the bladder [42]. Xie et al. described the identification of numerous T2Rs in the gastrointestinal tract [43], and they are also expressed in the brain [44], heart [45], respiratory tract [46], reproductive system [47], bone marrow stroma and the vascular wall [48]. T1R2/3 and T2Rs are believed to act antagonistically in regulating innate immunity in the respiratory tract [49]. Heterodimers of the taste receptors Taste receptor 1 and Taste receptor 3 (T1R1/3) regulate gastrointestinal hormones in the gastrointestinal tract [50]. It has also been reported that it controls the fertilization ability of sperm [51]. Salt-taste channels, epithelial sodium channels (ENaCs), are known to be expressed in the kidney, distal colon, bladder, stomach and lungs [52]. Almost every cell in the body expresses at least one member of the TRP channel family, which has distinct biophysical, mechanical, and pharmacological properties [53]. In particular, TRP channels involved in sensory signaling are the best characterized and have been used as experimental models to understand functional aspects.

Sensory receptors are located in different organs and are thought to have different functions, but many details remain unknown. In particular, it is not well understood how these sensory receptors in the gut react to the nutrients and phytochemicals in food, and how the subsequent signals from these food components are transmitted to the CNS. Recent reports suggest that food signals may contribute to homeostasis via sensory receptors in the gastrointestinal tract. For example, in the gastrointestinal tract, by the activation of GPCRs, sweet/umami/bitter taste receptors, lead to the secretion of gastrointestinal hormones by the intestinal secretory cells. These gastrointestinal (GI) hormones have been reported to activate the vagus nerve, and the stimulation is transmitted to the CNS, affecting eating and cognitive function [54,55]. As such reports accumulate, there is a growing need to investigate the importance of sensory nutrition, that is, how chemosensory signals from food contribute to maintaining homeostasis and reducing disease risk via the gastrointestinal-brain axis.

## 3. Bitter Taste Receptors and Polyphenols

### 3.1. Extra-Oral Bitter Taste Receptor and Gastrointestinal Hormones

Most pharmaceuticals are considered to present a bitter taste, and it is also known that bitter substances such as alkaloids and humulones are known to be present in plant food [56]. It has been well studied that these bitter substances are recognized by T2Rs, one of the G protein-coupled receptors (GPCRs), which are seven transmembrane receptors expressed within taste buds [57,58,59]. In humans, 25 members of the 291–334 amino acid long T2R superfamily are involved in the perception of a bitter taste [60,61]. As mentioned above, T2Rs are known to be distributed in various organs outside the oral cavity [43,44,45,46,47,48]. In recent years, research has increasingly focused on the relationship between bitter taste receptors expressed in the gastrointestinal tract and gastrointestinal hormones released by endocrine cells [62,63]. Typical gastrointestinal hormones incretins involved in these are glucose-dependent insulinotropic polypeptide (GIP), which is derived from K cells located in the upper small intestine, and glucagon-like peptides (GLP)-1, which are secreted mainly by L cells located in the distal small intestine [43,64,65]. Another hormone of particular interest is cholecystokinin (CCK), which is produced by I-cells in the upper part of the small intestine [66]. These hormones have been reported to be motility modulators of the gastropyloric duodenum that delay gastric content evacuation [66,67]. The peptides secreted by neuropodal cells inhibit the gastric inhibitory vagal motor circuit (GIVMC) via the vagus nerve and nucleus tractus solitarius (NTS), resulting in delayed gastric emptying. This mechanism is suggested to reduce appetite and thus energy intake [67,68]. 

In addition, the secreted incretin stimulates insulin secretion, which has been extensively studied for its ability to reduce the increase in blood glucose levels after a meal [68,69]. GLP-1 has glucagonostatic effects, and its analogue and receptor agonists are commonly used to treat type 2 diabetes [70,71,72]. GLP-1 receptor agonists, in particular, are widely used in clinical practice but are known to cause gastrointestinal complications. Therefore, it can be difficult to use this medicine as a preventative measure. On the other hand, GLP-1 secretion has been reported to be induced by oral administration of typical bitter substances such as quinine [73,74] and denatonium benzoate [75]. This has led to increased interest in drug discovery based on the interaction between bitter substances and T2Rs [76,77]. 

### 3.2. Interactions between Bitter Taste Receptors and Polyphenols

Polyphenols are chemically classified as flavonoids (Figure 1) including flavanols, flavanones, flavones, flavonols, isoflavones, anthocyanins, chalcones, lignans, tannins (proanthocyanidins, hydrolyzable tannins, fluorotannins, complex tannins) and other phenolic derivatives; refer to previous comprehensive reviews for the classification of polyphenols [19,78,79]. Polyphenol compounds are commonly associated with a bitter taste. However, there is limited practical evidence on the degree of bitterness associated with specific compounds as determined by human sensory evaluation. The specific sub-chemical structure of polyphenols that contributes to the bitter taste remains unclear. The interaction between polyphenols and T2Rs has been validated by in vitro experiments [80]. Most of the experiments have involved the expression of human T2Rs in HEK293 cells, the addition of polyphenols and the detection of calcium influx into the cells using fluorescent probes. A comprehensive survey of these previous studies was conducted using the following method.

The databases BitterDB (https://bitterdb.agri.huji.ac.il/dbbitter.php (accessed on 15 September 2023)), Phenol explore (http://phenol-explorer.eu/, accessed on 15 September 2023) and the database of natural TAS2R agonists (https://github.com/dipizio/Natural_TAS2R_agonists (accessed on 15 September 2023)) were used. Search formulae were “polyphenols”, “flavonoids”, “flavanols”, “flavanones”, “flavones”, “flavonols”, “flavanols”, “isoflavones”, “anthocyanins”, “chalcones”, “tannins”, “pro(antho)cyanidins”, “hydrolysable tannins”, “fluorotannins”, “complex tannins”, “phenols” and “taste receptor 2”, “T2R” and “TAS2R”. The survey was conducted in June 2023.

The results showed that 102 polyphenolic compounds were tested using the methods described above. The findings are summarized in Table 1. Most of the compounds interacted with T2R14 and T2R39. However, (−)-epicatechin, epigallocatechin gallate (EGCG) and procyanidins oligomers of epicatechin, specifically interacted with T2R5 [80]. Additionally, procyanidins were found to interact with T2R7 [80]. Castalagin, grandinin and vescalagin are hydrolyzable tannins that interact with T2R7. These tannins have a strong bitter taste similar to condensed tannin [80]. The results indicate that T2R7 may have a significant impact on the bitterness experienced by humans. Among these polyphenols, green tea’s EGCG and amarogentin, a type of secoiridoid glycoside, were found to interact with multiple T2Rs. EGCG was found to interact with T2R4, T2R5, T2R30 and T2R43 [3,80]. Amarogentin is a constituent of Gentian root, which has traditionally been used as a bitter stomachic. It has been reported to interact with eight T2Rs: T2R1, T2R4, T2R10, T2R30, T2R39, T2R43, T2R46 and T2R50 [81]. In these in vitro experiments, attempts have been made to calculate the EC_50_ of polyphenols that activate the T2R. The concentration range varied from a few micromoles to several tens of micromoles [3,4,81,82,83,84,85,86,87,88]. The EC_50_ values were similar to those of well-known bitter substances, such as absinthin, acetylthiourea, chloroquine, denatonium benzoate, ethylpyrazine and methylthiourea [58]. The above suggests that polyphenols have high potential as a seed for drug design using bitterness.

Furthermore, it should be noted that the in vitro experiments only targeted a limited number of the 25 T2Rs, therefore the interactions with all T2Rs remain unknown. Also, due to the different experimental conditions in each report, it was unclear to what extent the calculated EC_50_ was correct. The specific sub-chemical structure of polyphenols that contributes to the bitter taste remains unclear. The unclear steric structure of T2R was partly responsible for the uncertainty involved. In 2022, Xu et al. revealed the conformation of T2R46 for the first time using cryo-EM [89]. The paper showed that when the ligand strychnine binds to T2R46 in the apo state and activates the receptor, Y241 acts as a “toggle switch”. Meanwhile, DeepMind has developed an AI system so called “AlphaFold” which predicts the 3D structure of proteins, including an estimated structure of 25 T2Rs in humans “https://www.proteinatlas.org/search/tas2r” (accessed on 15 September 2023). Docking simulation methods and/or molecular dynamics simulation methods, etc., can be used to calculate the binding mode and binding energy of T2R and the ligand polyphenol based on available information. These methods would reveal the partial structure of polyphenols and their respective affinities for bitter taste receptors

## 4. Astringent Sensor and Polyphenols

### 4.1. Mechanisms of Recognition for Astringency Perception

Astringency is a taste that is specific to polyphenols, and bitter compounds can be either synthetic chemicals, such as pharmaceuticals and natural products. Astringency is a sensory attribute that is often described as a drying, roughening and puckering sensation in the mouth. Astringency is a common characteristic of anthocyanins, flavanols (including catechins, gallated catechins and procyanidins) and hydrolyzed tannins [5,90]. Furthermore, it is a crucial factor in determining the quality of foods and drinks such as berries, red wine and chocolate. However, the mechanisms by which astringent polyphenol stimuli are recognized remain unclear.

Taste refers to the sensation resulting from direct stimulation in taste cells in the taste buds. The taste sensation is transmitted centrally via the sensory branches of the facial nerve (VII), the glossopharyngeal nerve (IX) or the vagus nerve (X) [91]. In contrast, somatosensation is transmitted by the general sensory branches of the trigeminal (V), glossopharyngeal (IX) or vagus (X) nerves [92]. Astringency, rather than taste, has been reported to be transmitted as somatosensory information via the trigeminal nerve [93]. The mucosa surrounding taste buds contains various somatosensory receptors, such as mechanoreceptors, temperature receptors and nociceptors [94]. It is thought that humans perceive “flavor” through complex multisensory inputs in the oral cavity and olfactory inputs from the back of the nose that are transmitted to the CNS. Previous fMRI studies in humans have reported that the primary taste cortex, the insular cortex, is activated not only by sweet taste stimuli but also by somatosensory capsaicin and astringent stimuli, and that these three types of stimuli activate overlapping subregions [95]. Astringency has been reported to activate not only the insular cortex, a transient taste area, but also the superior orbitofrontal cortex, the cingulate cortex and frontal inferior triangularis with the intensity of activation being greater than that of sweet taste or capsaicin [95]. On the other hand, astringency was attributed to oral friction caused by the interaction of polyphenols with proline-rich proteins in saliva, which are recognized by mechanotropic receptors [5,6]. The conventional recognition hypothesis for astringent stimuli was explained by a three-stage model in which a reversible bond between the hydrophobic face of the aromatic ring of the polyphenol and the pyrrolidine ring of the proline residue of the protein first forms a soluble complex, additional polyphenols cross-link these peptides to form a larger insoluble complex, which then aggregates further [96]. However, it is not known whether such complex reactions take place in the oral cavity in the short time between food ingestion and swallowing. It has also recently been reported that interactions with mucins in saliva may also be involved in the formation of astringent sensations [97]. But the target molecules that recognize this phenomenon are not at all clear. According to this theory, salivary proteins or mucins are essential for mammals to recognize astringent stimulation. However, recent reports have shown that astringent polyphenols can activate GPCRs [93] or TRP channels [98,99,100] and trigger the activity of the trigeminal nerve independently of saliva and mucins. It is suggested that astringency perception is a type of somatosensory perception that does not necessarily require the presence of saliva or mucin.

### 4.2. Astringency Perception Is a Stressor

Consumption of cocoa extracts, which are rich in flavanols, a type of astringent polyphenol [92,96,98], can significantly reduce risk of cardiovascular diseases [16]. A significant reduction in the number of cardiovascular events and deaths from cardiovascular disease was observed when elderly people were given an average of 500 mg of flavanols per day for 3.6 years. Repeated flavanols are also known to improve blood pressure and risk factors for cardiovascular disease [101,102]. Furthermore, many intervention trials have demonstrated that two hours following a single intake of flavanol, there was a significant increase in blood flow-dependent vasorelaxation (FMD) levels [103]. However, the mechanism of the reproducible hemodynamic impacts of astringent polyphenols remains unknown.

There are also reports that there is an optimal dosage for this impact known as hormesis, i.e., this effect is attenuated at low or high doses [104,105,106] (Figure 2a). The hormetic concept is based on the idea that low levels of stress up-regulate adaptive responses that not only precondition, repair and restore normal function to damaged tissues/organs, but also modestly over-compensate, reducing ongoing background damage [107,108,109]. We have focused on the results of this study and developed an evaluation system to reproduce this effect in experimental animals [110]. A marked increase in blood flow in skeletal muscle arterioles was observed immediately after ingestion of the cocoa-derived flavanol fraction in rats [111]. In addition, it was found that at doses generally taken by humans from food (10 mg/kg), this blood flow-increasing effect was observed, but at doses 10 times higher, the effect disappeared [112]. Similar to humans, a hormesis response was observed in the hemodynamic effects of astringent polyphenols in experimental animals. It is widely recognized that exposure to stress in mammals leads to an increase in sympathetic nerve activity, causing alterations in the circulatory system [113,114]. Thus, several experiments were conducted using adrenergic receptor inhibitors to investigate the relationship between these effects and the sympathetic nervous system. According to the results, it was suggested that the rapid changes in hemodynamics after a single dose of flavanols were mediated by adrenergic receptors, i.e., sympathetic hyperactivation [112,115]. Furthermore, it has been suggested that the hemodynamic absence seen with high doses of flavanols compared to low doses was due to the activation of α2 receptors, an autoreceptor expressed in CNS [112,115]. Similarly, single doses of anthocyanins, polyphenols with an astringency, showed circulatory changes via sympathoadrenergic receptors [116]. Increased sympathetic activity is also known to promote white to beige fat cell conversion and skeletal muscle protein formation [117]. After being administered a flavanol fraction from cocoa for two weeks, browning was shown in mice inguinal adipose tissue. This was characterized by multilocular lipid droplets, a marked reduction in adipocyte size and increased expression of the heat-producing protein, uncoupling protein 1 (UCP-1) [118]. Repeated administration of the flavanol cinnamtannin A2 (A2, epicatechin tetramer) for two weeks also resulted in significant activation of the Akt/mTOR pathway and a marked increase in the mean muscle cross-sectional area in the soleus muscle [119] as similar to adrenalin agonist clenbuterol [120]. It has also been observed that after a single dose of flavanol, blood catecholamine levels were significantly increased [121] and urinary catecholamine excretion was also elevated [118]. These results confirm that FL induces sympathetic hyperactivity (Figure 2b).

Sympathetic nerve activity is known to be increased by exposure of stressors. Among food components, pungent components represented by capsaicin are known as stressors for mammals [122]. In addition, temperature changes [123], fasting [124] and exercise [125] are known to be stressors for mammals and can lead to longevity. It is known that when mammals are exposed to stress, activation of the hypothalamic-pituitary-adrenal axis (HPA) occurs along with hyperactivation of sympathetic nervous [126] (Figure 2b). To confirm that the effects of astringent polyphenols on circulatory dynamics are a stress response, we administered a single dose of flavanols and observed their effects on the HPA axis. The results showed a significant increase in the expression of the stress hormone corticotropin-releasing hormone (CRH) mRNA in the paraventricular nucleus of the mouse hypothalamus [127,128]. It was also found that the duration of CRH mRNA expression onset shortened as the doses increased. A subsequent increase in blood cortisol (or corticosterone in rodents) levels was also confirmed. The administration of astringent polyphenols leads to hyperactivation of sympathetic nervous and activation of the HPA axis in mammals. These results indicate that astringency is perceived as a stressor like capsaicin.

### 4.3. Astringent Polyphenols and TRP Channels

As mentioned in the previous section, astringency is likely to be a somatic sensation as with other stressors [93,94]. Schöbel et al. reported that the sensation of astringency is not impaired in human subjects when the taste nerve is denervated or blocked by local anesthesia. In addition, they showed that salty, sweet, sour and bitter tastes were almost completely lost in these subjects, and that subjects lost the sensation of astringency only when both trigeminal and gustatory nerves were blocked [93]. In 1997, it was demonstrated that the somatic sensation caused by capsaicin stimulation is recognized by the transient receptor potential vanilloid 1 (TRPV1) channel, which is one of the nociceptors [129]. In mammals, there are 28 different TRP channel proteins expressed, which are classified into seven subfamilies based on amino acid sequence homology [53]. TRP channels are involved in a variety of sensory responses, including heat, cold, pain, stress, vision and taste, and are activated by a number of stimuli. These channels are composed of six transmembrane polypeptide subunits that assemble as a tetramer to form a cation-permeable pore [130]. TRPV1 and transient receptor ankyrin 1 (TRPA1) are expressed mainly in sensory neurons and are also distributed in the oral and gastrointestinal tracts [131,132]. It is widely recognized that capsaicin activates sympathoexcitation via the CNS by activating TRPV1 expressed on sensory nerves, thereby promoting heat production [133] and altering hemodynamics [134]. The astringent polyphenol flavanols also have been observed to have the identical effect described above. Interactions between TRP channels and astringent polyphenols have been reported in several cellular experiments [98,99,100]. In Ca^2+^ imaging experiments, it has been reported that EGCG, an astringent and bitter polyphenol, activates the mouse enteroendocrine cell line STC-1. TRPA1 receptor inhibitors canceled this effect [99]. We investigated whether astringent polyphenols activate TRP channels and investigated the involvement of TRP channels in the effect of increasing blood flow in rats. The rats were administered flavanol A2, and TRPV1 or TRPA1 channel inhibitors simultaneously through single oral doses. The marked increase in blood flow observed with A2 alone was eliminated by the combined use of TRP channel inhibitors [135]. These findings suggest that TRP channels contribute to the sympathetic hyperactivity of flavanols as a stress response. The interaction between A2 and each TRP channel was then observed using modeling simulation methods. The binding energies of the A2 and ligand binding sites for each TRP channel were markedly higher than that of the ligand [135]. These results suggested that there was no direct interaction between them.

Polyphenols are stored in vacuoles within plants, which have a weakly acidic pH of 4–5, and are stable [136]. However, when consumed by mammals, they are exposed to neutral pH conditions in the gut such as the mouth, small intestine and large intestine. It is widely recognized that flavanols [137] and anthocyanins [138], which are astringent polyphenols, undergo rapid oxidation and decomposition or condensation under neutral pH. In our study, (−)-epicatechin (EC) and its tetramer, A2, also showed superoxide radical (O_2_^•−^) scavenging activity at acidic pH but significantly enhanced O_2_^•−^ production under pH 7 [135]. Furthermore, an in vitro study conducted on TRP channel-expressing HEK293 cells has demonstrated that oxidized EGCG, which was incubated at neutral pH for several hours, significantly enhances calcium influx [100]. In contrast, freshly dissolved, non-oxidized EGCG showed no such effect. In addition, peripheral vascular blood flow in rats was not increased after co-administration of O_2_^•−^ scavengers and A2, the blood flow increasing effect observed with A2 alone was nullified [135]. 

It was reported that TRPV1 and TRPA1 can be activated by reactive oxygen species (ROS) through cytoplasmic side cysteine residues [139]. In hTRPV1, Cys-258 is highly oxidizable, leading to the formation of a disulfide bond between Cys-258 and Cys-742, which opens the channel [140]. In TRPA1, mutational analysis has reported the presence of multiple reactive oxygen species domains, including Cys421, Cys621, Cys641 and Cys665 [141]. These findings suggest that orally ingested flavanols and anthocyanins produce ROS in the neutral pH environment of the gastrointestinal tract, which are oxidatively degraded by successive oxidative reactions in the molecule, and the produced ROS may interact with the ROS recognition domain of the TRP channels and become activated (Figure 3). Furthermore, the responses of ROS-TRP channels on sensory nerves could be perceived in the CNS as an astringent stimulation. However, further research is needed on how TRP channels are involved in the mechanism of astringecyperception.

## 5. Bioavailability of Polyphenols 

Numerous studies have been conducted on the bioavailability of polyphenols [142,143,144]. Although a proportion of ingested polyphenols are absorbed in the small intestine, the rate and total amount of absorption of polyphenols are known to be strongly influenced by their chemical structure. The bioavailability in humans, the urinary excretion of most polyphenols is a few percent, whereas the excretion of astringent polyphenols such as anthocyanins and flavanols is quite low [145]. Flavonoids (Figure 1) as one of the main polyphenols are compounds with a 3,4-dihydro-2-phenyl-2H-1-benzopyran structure and there might be an inverse correlation between the number of hydroxyl groups in the B-ring and bioavailability. For example, in an intervention study, the urinary excretion rate of pelargonidin, an anthocyanin aglycon with a single hydroxyl group on the B-ring, was reported to be 1.8% [146]. In contrast, cyanidin, which has two hydroxyl groups, is around 0.1% [147]. Methylated flavonoids, which have methoxy groups on the B ring, are also known to be highly absorbable [148]. These differences in absorption may depend on the chemical stability of polyphenols in the neutral to slightly alkaline conditions of the small intestine. For example, cyanididins are readily degraded under neutral pH conditions to protocatechuic acid (PCA) and phloroglucinaldehyde (PGA), and it has been reported that blood concentrations of the degradation product PGA were four times higher than those of cyanidin [149]. Most polyphenolic compounds are known to be glucuronidated or sulfated by the small intestine or liver and exist as metabolites in the blood [18]. In addition, the metabolites of polyphenols in plasma have difficulty penetrating cell membranes and BBB because they changed more water soluble than parent compound. Ingested polyphenols pass through the upper gastrointestinal tract and reach the large intestine. A part of ingested polyphenols is known to be degraded by intestinal bacteria in the colon. Several comprehensive reviews of these have been reported so far [20,150,151]. However, the amount and type of metabolic degradation of polyphenols by gut bacteria relative to the total polyphenols consumed, and the kinetics of this degradation, is not well understood. Unlike nutrients and easily absorbed chemicals, such as fat-soluble substances, ingested polyphenols are present in high concentrations in the oral cavity, stomach, small and large intestines. Therefore, until they are excreted from the body in feces, they keep in contact with I-, K- and L-cells that express bitter taste receptors, or with gastrointestinal sensory nerves and epithelial cells that express TRP channels for an extended period. Considering these behaviors in the gastrointestinal tract, it is likely that polyphenols exert their various physiological effects via sensory receptors expressed in the gastrointestinal tract.

## 6. Biological Regulation through Bitterness and Astringency of Polyphenols

Polyphenols have been shown in epidemiological studies to have the potential to prevent cardiovascular disease, neurodegenerative diseases and age-related sensory organ deterioration [10,11,12,13]. Large-scale intervention trials of flavanols have also reported reduced cardiovascular mortality [16] and hippocampus-dependent improvements in cognitive function [17]. As previously discussed, the nervous system, including CNS, plays a significant role in controlling obesity, hyperglycemia, dyslipidemia and hypertension, all of which are risk factors for cardiovascular disease [152]. However, it is important to note that polyphenols are poorly absorbed and, even if absorbed, are only present in the blood as metabolites. They are unlikely to pass through the BBB and be distributed in the brain. Furthermore, there is no discussion of how bitterness and astringency contribute to these effects. In considering the mechanisms of these beneficial effects of polyphenols, we proposed to discuss the involvement of afferent and efferent nerve. Therefore, the following is an overview of the various physiological effects of polyphenols reported so far and how they contribute to bitter and astringent perception.

### 6.1. Circulation and Polyphenol

The effects of cocoa astringent flavanol on the circulatory system are well studied: a meta-analysis of RCTs of adult participants at cardiometabolic risk reported between 2000 and 2021 observed a marked increase in FMD [153]. These activities on FMD of astringent anthocyanin [154] was also confirmed by the meta-analyses. On the other hand, for compounds without a pronounced taste, such as resveratrol [155] or isoflavones [156], the onset of action has been reported to depend on the study design. In addition, there are few reports of other polyphenol enhancement in FMD. This is because it is difficult to identify the main compounds responsible for biological activity because many experiments have conducted mixtures. The FMD-enhancing effect of a single dose of flavanols shows a non-linear hormesis response [105,106]. Interestingly, anthocyanins [104,154] or flavanol [104] have been reported to be more effective on FMD in single oral doses compared with repeated doses. In addition, repeated intake of these polyphenols is also well known to affect blood pressure [157]. While meta-analyses have shown that the repeated intake of flavanols significantly reduces systolic and diastolic blood pressure [18,158]. 

In experimental animal studies, a single oral administration of flavanols or anthocyanins also results in a significant increase in blood flow of skeletal muscle arteriole [111,112,116]. Furthermore, repeated administration of these chemicals considerably decreased the mean blood pressure in rodents [159,160]. These modifications are consistent with the changes that occur in the circulatory system as a result of exercise (Figure 4). In brief, acute exercise results in temporary alterations in the circulatory system, including elevated blood pressure and heart rate, as well as increased microvascular blood flow; whereas repeated exercise causes a decrease in blood pressure [158]. Acute exercise increases skeletal muscle blood flow through hyperactivation of the sympathetic nervous system, subsequently inducing shear stress on vascular endothelial cells [161]. Mechanosensors consisting of vascular endothelial growth factor (VEGF) receptor 2, vascular endothelial (VE) cadherin and CD31 (platelet endothelial cell adhesion molecule-1, PECAM-1) expressed on endothelial cells respond to shear stress and release vasorelaxing factor NO via the Akt/eNOS pathway, causing the vessel to relax [162] (Figure 4a). The rise in FMD resulting from astringent polyphenols observed in the intervention trials may be attributed to these alterations in hemodynamics. In addition, these alterations lead to the differentiation of vascular endothelial cells via mechanosensors, with the repetition of the exercise, these changes in hemodynamics are known to trigger the formation of new capillaries [163]. Continuous fluid shear stress induces angiogenesis through the cellular production of proangiogenic growth factors, such as VEGF. Resting endothelial cells are transformed into tip cells by the secretion of angiogenesis-promoting factors from a hypoxic or metabolically active microenvironment [164]. These tip cells proliferate to become stalk cells, which extend a growing sprout and form a lumen. Once the two sprouts fuse, they form a capillary tube, which lowers the resistance to blood flow and the endothelial cells return to quiescence. In this quiescent state, endothelial cells produce mechano-sensing complexes and form tight junctions [165] (Figure 4b). These factors lead to angiogenesis, which is an increase in blood vessel volume that, in turn, results in a decrease in blood pressure.

In rodent studies, a single administration of astringent polyphenols induced phosphorylation of eNOS in the aorta [111,112,115,116,160]. Moreover, repeated administration of astringent polyphenols to rodents has been shown to lower blood pressure [159,160]. Furthermore, repeated administration reduced blood pressure and significantly increased CD31 expression, a marker of neovascularization, in the soleus [160]. This change suggested the promotion of angiogenesis. This process of angiogenesis increases vascular capacity, reducing the risk of exposure to shear stress. Therefore, repeated ingestion of astringent polyphenols resulted in a decrease reduction in the FMD response along with blood pressure.

### 6.2. Blood Glucose and Polyphenols

Interest in using bitter compounds to improve glucose tolerance by stimulating gastrointestinal hormone secretion has recently increased [76,77]. Most polyphenols have a bitter taste and low bioavailability, which makes them potentially useful for this objective. For example, interventional studies have been conducted on changes in glucose tolerance and blood levels of incretin after single doses of green tea flavanol [166], coffee chlorogenic acid [167,168] or combinations thereof [166], olive oleuropein [169] and anthocyanin fraction from blackcurrant [170] in healthy subjects. Intake of 540 mg of green tea flavanol reduced the increase in blood glucose levels caused by a high-fat, high-carbohydrate diet, increased blood levels of GLP-1 and reduced GIP levels [166]. In addition, when 355 mg of chlorogenic acid was given to healthy subjects, an increase in blood levels of GLP-1 was only observed in subjects with a low insulinogenic index [168]. These combinations (beverages containing 540 mg of green tea flavanols and 150 mg of chlorogenic acid) have also been reported to reduce the rise in blood glucose, increase blood GLP-1 levels and reduce GIP levels after consumption of high-fat, high-carbohydrate cookies [166]. A single oral dose of oleuropein (20 mg) decreased blood glucose and dipeptidyl peptidase (DPP)-4 levels, which inactivates active GLP-1, and increased blood levels of GLP-1 and insulin two hours after consumption of the test meal [169]. These findings suggest that a single ingestion of bitter polyphenols may increase incretin secretion from gastrointestinal secretory cells, promote active incretin retention, induce insulin secretion and suppress meal-derived blood glucose levels after a few hours. Improvements in glucose tolerance as measured by the homeostatic model assessment for insulin resistance (HOMA-IR) have been reported in studies investigating repeated administration of green tea flavanol, either alone [171] or in combination with chlorogenic acid [172], pine bark flavanol [173], acacia flavanols [174], red wine polyphenol [175], eriocitrin [176] and curcumin [177]. However, studies with resveratrol have reported both improvement [178] and no improvement in insulin resistance [179]. Further research is needed to determine the specific types and amounts of polyphenols that are effective in improving glucose tolerance. Shamsudin et al. showed that the antidiabetic effect can depend on the chemical structure of the flavonoid [180]. It was reported that flavonoids with antidiabetic activity should have a C2–C3 double bond (C ring) in both the A and B rings of the flavonoid skeleton and hydroxyl groups in the C3′, C4′, C5 and C7 positions (Figure 1). The result of the study demonstrate that the beneficial activity of polyphenols is highly variable, depending on the chemical structure. Therefore, studies that elucidate the subchemical structures of polyphenols essential for T2R activation are crucial for drug discovery.

### 6.3. Obesity and Polyphenols

It has been suggested that consuming polyphenols may help to control obesity [181,182]. A meta-analysis of clinical trials published between 2010 and 2021 showed that polyphenols produced statistically significant reductions in body weight, BMI and abdominal circumference, but no significant reduction in body fat. Subgroup analyses showed that polyphenol intake had significant effects in subjects under 50 years of age, in the Asian population and in patients with obesity-related health problems, for more than three months and at doses of around 220 mg per day [183]. To date, the anti-obesity effects of polyphenols have been attributed to (1) appetite suppression; (2) reduced digestion and absorption of lipids and carbohydrates through inhibition of digestive enzymes; (3) adipocyte differentiation; (4) regulation of lipid metabolism; (5) stimulation of energy expenditure; (6) improvement of intestinal microflora; (7) amelioration of obesity-related mild inflammation; and (8) reduction of oxidative stress [184,185,186]. However, it should be noted that the bioavailability of polyphenols is very low. While they may inhibit digestive enzymes and improve gut microbiota, other effects are likely to be secondary. As mentioned above, bitter substances such as polyphenols may reduce energy intake due to suppressed appetite via the secretion of gastrointestinal hormones, resulting in vagus nervous stimulation [43,67]. Polyphenols appear to have a similar effect to GLP-1 agonists in weight loss by suppressing appetite and reducing gastric emptying [187,188]. Increased sympathetic activity after ingestion of astringent polyphenols is thought to cause fat browning, as confirmed in the study using rodents [118]. In addition, this promotion of lipid oxidation induced by the consumption of astringent polyphenols has also been shown to increase HDL cholesterol levels in the blood [189,190]. Many of the hypotheses proposed as mechanisms for the anti-obesity effects of polyphenols can be explained by their bitter or astringency.

### 6.4. Brain Function and Polyphenol

Gastrointestinal hormones secreted by the ingestion of polyphenols have been reported to activate the vagus nerve as a neurotransmitter [191]. Gastrointestinal hormones are secreted by neuropodal cells in response to bitter, sweet and umami taste stimuli, and this sensory information is transmitted directly to the NTS within milliseconds [192,193,194]. Upstream of the NTS, it projects to multiple brain regions, and transsynaptically to the dorsal hippocampus [195] and frontal cortex [196]. These sensory transmissions have been shown to improve mood and memory in studies using severed gastrointestinal vagus nerves [192]. Recently, results from intervention studies have shown that stimulating the vagus nerve can help depression recovery [197]. The bitterness of polyphenols may play a role in the homeostasis of the brain. 

Polyphenols with a markedly astringency have been shown to trigger the stress response and to activate the hypothalamic-pituitary-adrenal (HPA) axis with an increase in sympathetic activity [127,128] as well as the other stressors like capsaicin. Stress causes CRH to be secreted from the paraventricular nucleus of the hypothalamus into the pituitary portal system, stimulating the anterior pituitary to release adrenocorticotropic hormone (ACTH) [198]. ACTH subsequently circulates to the adrenal glands and stimulates the release of cortisol into the blood [199]. Cortisol readily crosses BBB and binds to receptors in the amygdala, prefrontal cortex and hippocampus [200] (Figure 2). Chronic stress, such as fear and anxiety, is an allosteric load and excess cortisol is released by the adrenal glands [201]. The resulting prolonged activation of neuroendocrine, cardiovascular and emotional responses can be damaging to health with increased cardiovascular risk, cognitive dysfunction and depressive mood [202]. Although exercise is a stressor that activates the HPA axis and increases blood cortisol levels, it has effects on cognition and mood, and promotes neurogenesis in adults [203,204]. A cortisol paradox between chronic stress and exercise on brain function has been well observed [205,206,207]. A comparable occurrence has been noted with astringent polyphenols. The reason for this paradox is unclear, but one possibility is the involvement of cortisol-glucocorticoid receptor (GR)-dopamine (DA)-dopamine D2 receptors [208]; cortisol secreted into the blood activates GR in the medial prefrontal cortex and enhances glutamatergic input from mPFC to ventral tegmental area (VTA). This input to the VTA enhances the projection of DA neurons from the VTA to the mPFC. This increase in DA in the mPFC activates GABAergic neurons in the nucleus of the anterior nucleus of the terminal line (aBNST), which project to the paraventricular nucleus (PVH) of the hypothalamus. It provides negative feedback control and prevents over-reaction of the HPA axis [208]. These mechanisms suggest that exercise or astringent polyphenols may have the effect of reducing excessive HPA activation and improving mood and memory function, despite increasing basal levels of glucocorticoids. 

In a large intervention study in elderly subjects, 3.6 years of cocoa extract, mainly composed of astringent polyphenols, was reported to restore hippocampus-dependent memory in participants in the lower tertile of habitual diet quality or flavanol intake [17]. In rodents, repeated administration of highly astringent catechin tetramers A2 has also been shown to improve spatial memory with enhanced adult neurogenesis [209]. In summary, astringent stimulation may benefit brain function via the GR-DA-DAR pathway as well as exercise, but much more research is needed to elucidate the mechanism.

## 7. Conclusions

The consumption of polyphenols is beneficial for the maintenance and promotion of human health. However, the mechanisms remain unclear. As most of polyphenols have a bitter taste, migrate from the oral cavity to the gastrointestinal tract and interact with intestinal secretory cells, they likely regulate sugar metabolism or feeding via the T2Rs. On the other hand, if we consider the effects of astringent polyphenols on the circulatory system, metabolism and brain function, their effects have a great deal in common with the benefits of exercise. Astringency, a stressor, elicits a hormetic response in sympathetic nerve overactivity and is considered to have beneficial effects in moderate doses. Research on the bio-modulation of polyphenols with taste, which has not received much attention to date, may provide a solution to the polyphenol paradox, in which polyphenols exert bio-regulatory effects despite their extremely low bioavailability.

## Figures and Tables

**Figure 1 biomolecules-14-00234-f001:**
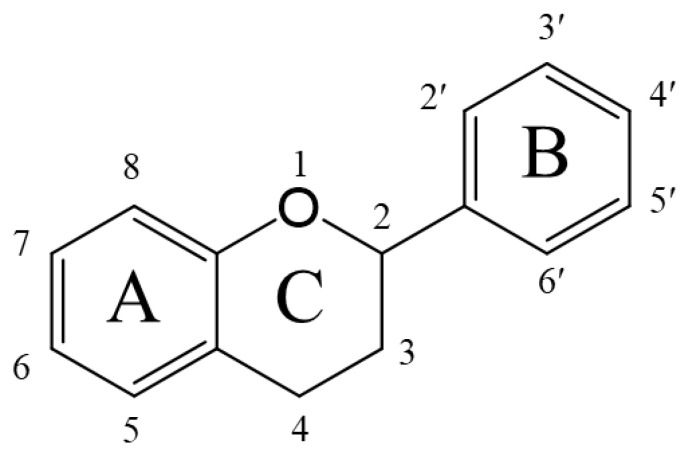
Chemical structure of flavonoid.

**Figure 2 biomolecules-14-00234-f002:**
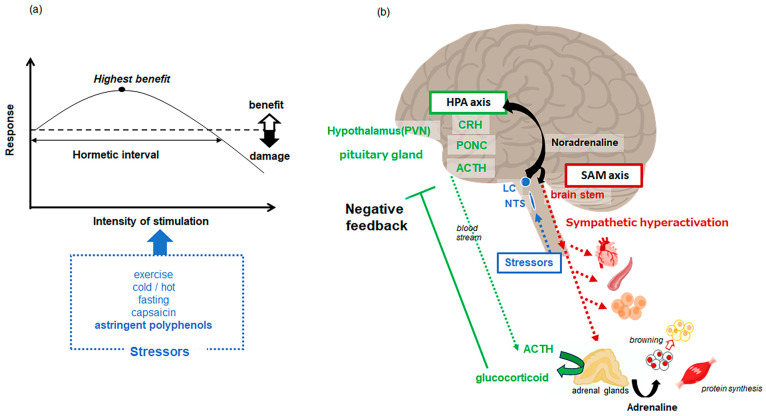
Hromesis (**a**) and the mechanism of stress response (**b**) induced by astringent polyphenols. Stress stimuli input to the nucleus fasciculus solitarius (NTS) project norepinephrine via the locus coeruleus (NTS) to the hypothalamus, brainstem and other brain regions. The stimulation increases sympathetic nerve activity (sympathetic nervous -adrenal medulla axis), which causes changes in various organs via adrenergic receptors; adrenaline is also secreted into the blood from the adrenal glands, inducing browning in adipose and synthesizing skeletal muscle proteins. The paraventricular nucleus (PVN) of the hypothalamus synthesizes corticotropin-releasing hormone (CRH), which stimulates the secretion of adrenocorticotropic hormone (ACTH) from proopiomelanocortin (PONC) in the pituitary gland into the blood. ACTH promotes the secretion of glucocorticoids from the adrenal glands into the blood, and glucocorticoids provide negative feedback to the hypothalamus-pituitary-adrenocortical (HPA) axis via receptors expressed CNS.

**Figure 3 biomolecules-14-00234-f003:**
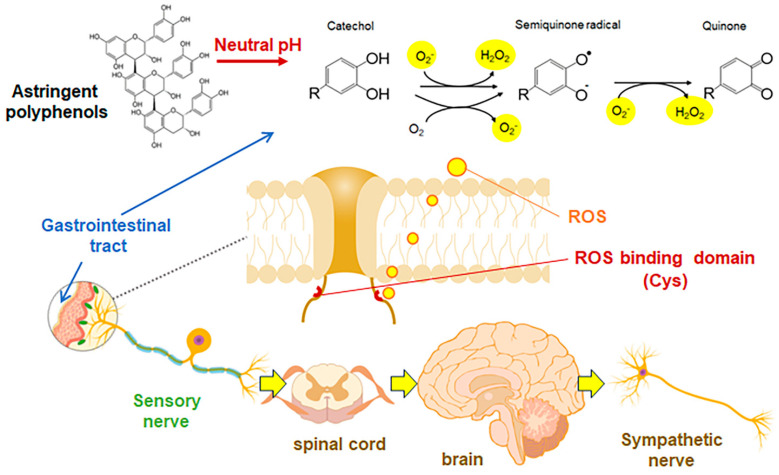
Hypothetical recognition mechanism of astringent polyphenols by TRP channel.

**Figure 4 biomolecules-14-00234-f004:**
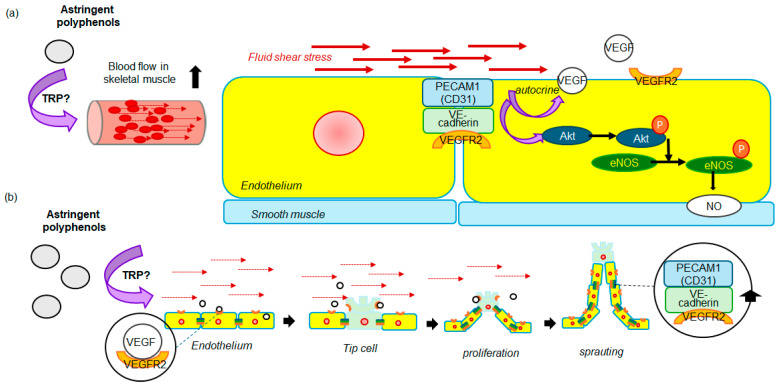
Illustrations of two hypotheses for the mechanisms underlying the observed effects of a single (**a**) and repeated oral administration (**b**) of astringent polyphenols on micro-and systemic circulation.

**Table 1 biomolecules-14-00234-t001:** The interaction of polyphenols and T2Rs in vivo studies.

T2R	Compounds
1	Isoxanthohumol (85) Resveratrol (60) Xanthohumol (85) Amarogentin (87)
3	ND
4	(−)-Epicatechin (3) Epigallocatechin gallate (4) Amarogentin (87)
5	(−)-Epicatechin (3) Epigallocatechin gallate (4) Procyanidin B1 (4) Procyanidin B2g (4) Procyanidin B4 (4) Procyanidin B7 (4) Procyanidin C2 (3) PGG(pentagalloylglucose) (3) Punicalagin (4)
7	Procyanidin B1 (4) Malvidin-3-glucoside (3) Castalagin (4) Grandinin (4) Punicalagin (4) Vescalagin (4)
8	ND
9	ND
10	Amarogentin (87)
13	ND
60	(+)-Catechin (86) (−)-Epicatechin gallate (84) Epigallocatechin gallate (84) Eriodictyol (86) Flavanone (86) Hesperitin (86) Homoeriodictyol (86) Liquiritigenin (86) Naringenin (86) 8-Prenylnaringenin (85) Pinocembrin (86) Luteolin (86) Nobiletin (84) Scutellarein (86) Apigenin (86) Chrysin (86) Chrysoeriol (86) Datiscetin (86) 5,4-dihydroxyflavone (86) 6,4-dihydroxyflavone (86) 5,7,2-trihydroxyflavone (86) 3,7,4-trihydroxyflavone (86) 3,6,3,4-tetrahydroxyflavone (86) 5,7-dimethoxyflavone (86) 6,7-dimethoxyflavone (86) 5,7,4-trimethoxyflavone (86) 4-hydroxy-6-methoxyflavone (86) 6-methoxyflavone (86) 7,4′-dihydroxyflavone (86) 6-methoxyluteolin (86) Flavone (86) Herbacetin (86) Isorhamnetin (86) Kaempferol (86) Morin (86) Myricetin (86) Quercetagetin (86) Quercetin (85) Taxifolin (86) Fustin (86) (±)-equol (82) Biochanin A (86) Daidzein (86) Formononetin (82) Genistein (82) Glycitein (82) Isoflavone (82) Prunetin (82) 7-hydroxyisoflavone (82) 7,3,4-trihydroxyisoflavone (82) 6,7,4-trihydroxyisoflavone (82) 7,8,4′-trihydroxyisoflavone (82) 7,4-dimethoxyisoflavone (86) Cyanidin chloride (86) Pelargoninidin chloride (86) Isoxanthohumol (85) Tangeretin (84) 4-hydroxychalcone (86) 2,2′,4′-trihydroxychalcone (86) 3,2-dihydroxychalcone (86) 4,2,5-trihydroxychalcone (86) Butein (86) Chalcone (86) Eriodictyol chalcone (86) Isoliquiritigenin (86) Resveratrol (86) Xanthohumol (85) Procatechuic acid (4) Ferulic acid (4) Vannillic acid (4) Coumestrol (82) Sulfuretin (86) Umbelliferone (60) Silibinin (86)
16	Arbutin (88) catechole (88)
19	ND
20	ND
30	Epigallocatechin gallate (4) Procatechuic acid (4) Amarogentin (87)
31	ND
38	ND
39	(+)-Catechin (86) (−)-Epicatechin (3) (−)-Epicatechin gallate (83) (−)-Epigallocatechin (4) Epigallocatechin gallate (4) Procyanidin B2g (4) Eriodictyol (86) Hesperitin (86) Homoeriodictyol (86) Liquiritigenin (86) Naringenin (86) Pinocembrin (86) Genkwanin (86) Luteolin (86) Scutellarein (86) Tricetin (86) Apigenin (86) Chrysin (86) Datiscetin (86) 4-hydroxyflavone (86) 5,2-dihydroxyflavone (86) 5,4-dihydroxyflavone (86) 6,4-dihydroxyflavone (86) 5,7,2-trihydroxyflavone (86) 3,7,4-trihydroxyflavone (86) 3,6,3,4-tetrahydroxyflavone (86) 5,7-dimethoxyflavone (86) 6-methoxyflavone (86) 7,4-dihydroxyflavone (86) 6-methoxyluteolin (86) Flavone (86) Fisetin (86) Gossypetin (86) Herbacetin (86) Isorhamnetin (86) Kaempferol (86) Morin (86) Myricetin (86) Quercetagetin (86) Taxifolin (86) Fustin (86) (±)-equol (82) acetylgenistin (82) Biochanin A (86) Daidzein (86) Formononetin (82) Genistein (82) Genistin (82) Glycitein (82) Glycitin (82) Malonylgenistin (82) 7-hydroxyisoflavone (82) 7,3,4-trihydroxyisoflavone (82) 6,7,4-trihydroxyisoflavone (82) 7,8,4-trihydroxyisoflavone (82) Cyanidin chloride (86) Pelargoninidin chloride (86) 2,2′,4′-trihydroxychalcone (86) 3,2-dihydroxychalcone (86) 4,2,5′-trihydroxychalcone (86) Butein (86) Chalcone (86) Eriodictyol chalcone (86) Isoliquiritigenin (86) Resveratrol (86) PGG(pentagalloylglucose) (3) Amarogentin (87) Coumestrol (82) Sulfuretin (86) Silibinin (86)
40	Isoxanthohumol (85) Xanthohumol (85)
41	ND
42	ND
43	Epigallocatechin gallate (4) Amarogentin (87)
45	ND
4	Nobiletin (89) Tangeretin (89) Amarogentin (87)
50	Amarogentin (81)
60	ND

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
