# Peer review of "Sensory Nutrition and Bitterness and Astringency of Polyphenols"

_biomolecules, 2024, doi:10.3390/biom14020234_

Round 1

Reviewer 1 Report

Comments and Suggestions for Authors

The article deals with a topic on which many reviews have been published recently. There are significant errors in the introduction (lines 55-65) and along the paper. The bitter taste receptors are 25  so they have to be indicated as T2Rs. Line 521 analogues should be corrected in agonists. Some references are quite old (3,9,17), just to quote some

Author Response

Thank you for conveying comments and reviewers’s suggestions. We appreciated the suggestions and revised the manuscript accordingly.

RESPONSE TO REVIEWER 1 : We wish to express our appreciation to Reviewer 1

Comment 1

 There are significant errors in the introduction (lines 55-65) and along the paper.

- Response

We have carefully revised L55-65. Please see text for details

Comment 2

 The bitter taste receptors are 25  so they have to be indicated as T2Rs.

-Response

We have carefully revised.

Comment 3

 Line 521 analogues should be corrected in agonists.

-Response

We have revised.

Comment 4

 Some references are quite old (3,9,17), just to quote some

-Response

We have carefully revised.

Reviewer 2 Report

Comments and Suggestions for Authors

1The topic and the hypothesis elaborated by the authors are quite interesting, but the paper require major improvement before being publishable.

11)    The paper needs more coherence and structure, concordance between the subtitles and the content of the subsection. The paper also lack synthesis of information.

22)    Regarding the statement: “ Taste receptors respond to salts, sugars, amino acids, alkaloids, acids and fats[8, 9].”, the author should know that there are many other classes of compounds that activate taste receptors, e.g. flavone, polyphenols, see references cited in the paper in other places

                Meyerhof, W.; Batram, C.; Kuhn, C.; Brockhoff, A.; Chudoba, E.; Bufe, B.; Appendino, G.; Behrens, M., The molecular receptive 776 ranges of human TAS2R bitter taste receptors. Chem Senses 2010, 35, (2), 157-70.

Soares S, Silva MS, García-Estevez I, Groβmann P, Brás N, Brandão E, Mateus N, de Freitas V, Behrens M, Meyerhof W. Human Bitter Taste Receptors Are Activated by Different Classes of Polyphenols. J Agric Food Chem. 2018 Aug 22;66(33):8814-8823

33)    Similarly, TRPs are activated by many volatile compounds, not only menthol. For instance see ref Kim H, Kim M, Jang Y. Inhaled Volatile Molecules-Responsive TRP Channels as Non-Olfactory Receptors. Biomol Ther (Seoul). 2023 Aug 8. Please reformulate.

44)    There several scientifically confusing statements that need to be reformulated:

For example, in the gastrointestinal tract, by the activation of GPCRs, olfactory receptors and sweet/umami/bitter taste receptors, secrete gastrointestinal hormones. Taste receptors do not secrete hormones!

2.1. Bitter Taste Receptors Expressed Extra-Oral Cavity and GI Hormones” What does it mean?! What is “Extra-Oral Cavity”??

Regarding the subtitles “3. Astringent Taste Receptors and Polyphenols  3.1. Mechanisms of Astringent Taste Perception” astringency is not recognized as a taste, it is considered a trigeminal mediated chemosensation!

55)    Regarding the statement: “Polyphenols are chemically classified as flavonoids (Figure. 1a) including flavanols, 151 flavanones, flavones, flavonols, flavanols, isoflavones, anthocyanins), chalcones, tannins 152 (proanthocyanidins, hydrolysable tannins, fluorotannins, complex tannins) and phenols 153 [55, 80].” Polyphenols are not only flavonoids, chalcones, tannins and phenols! There are several other classes of polyphenols, besides these : phenolic acids, polyphenolic amides, stilbenes, lignans, etc. Please see for ref Tsao R. Chemistry and biochemistry of dietary polyphenols. Nutrients. 2010 Dec;2(12):1231-46

66)    Regarding Figure 1, the authors should present the basic chemical structure of all the classes of polyphenols (not only flavonoids), or no structures at all.

77)    Some statements lack references. e.g. “flavanols, a type of astringent polyphenol,

88)    I do not understand why the a subsection about polyphenols and stress response in the section entitled “Astringent Taste Receptors and Polyphenols”? This shows a lack of focus, and has nothing to do with the taste receptors and the title…

99)    There is redundant information. For instance in subsection 3.3. “According to the research results to date, it seems that astringency is likely to be a 306 somatic sensation [94, 95].” The authors have already talked about this in a previous subsection.

110) Why talk about capsaicin, a pungent amide alkaloid in a paper about polyphenols, in a section about astringent polyphenols?

111) Biological Regulation through Bitter and Astringency of Polyphenols” should be “Biological Regulation through Bitterness and Astringency of Polyphenols

12) Abbreviations should be explained at their first appearance in the text. e.g. FMD

113) The connection between the two following statement should be imporved, because it is not clear for the reader the idea of the authors that astringency as a stress acts through similar mechanism with those initiated by exercise.

114) Reformulate “polyphenols…migrate” in Conclusion section.

Author Response

Thank you for conveying comments and reviewers’s suggestions. We appreciated the suggestions and revised the manuscript accordingly.

RESPONSE TO REVIEWER 2 :We appreciated the suggestions and revised the manuscript accordingly.

Comment 1

Regarding the statement: “Taste receptors respond to salts, sugars, amino acids, alkaloids, acids and fats[8, 9].”, the author should know that there are many other classes of compounds that activate taste receptors, e.g. flavone, polyphenols, see references cited in the paper in other places.

-Response

We revised as follows;

Taste receptors respond to salts, sugars or sweeteners, amino acids, alkaloids, various bitter chemicals such as cycloheximide propylthiouracil denatonium benzoate, strychnine, etc., acids and fats.

Comment 2

Similarly, TRPs are activated by many volatile compounds, not only menthol.

-Response

We revised as follows and added the cited papers.

In addition, stimuli such as temperature, osmotic pressure, pH, pungency and many volatile compounds such as menthol aldehyde relatives, anesthetics, nicotine etc., are recognized by transient receptor potential (TRP) channels [30-33].

  1. Kim, H.; Kim, M.; Jang, Y., Inhaled Volatile Molecules-Responsive TRP Channels as Non-Olfactory Receptors. Biomol Ther (Seoul) 2023.

Comment 3

There several scientifically confusing statements that need to be reformulated:“For example, in the gastrointestinal tract, by the activation of GPCRs, olfactory receptors and sweet/umami/bitter taste receptors, secrete gastrointestinal hormones.”  Taste receptors do not secrete hormones!

- Response

We revised as follows;

For example, in the gastrointestinal tract, by the activation of GPCRs, olfactory receptors and sweet/umami/bitter taste receptors, secrete gastrointestinal hormones from intestinal secretory cells.

Comment 4

2.1. Bitter Taste Receptors Expressed Extra-Oral Cavity and GI Hormones” What does it mean?! What is “Extra-Oral Cavity”??

- Response

The term "Extra-oral cavity" is frequently used in recent papers regarding taste receptors expressed in gut other than the oral cavity. for example, Ziegler F, et al. doi: 10.1038/s42003-023-04971-3, Behrens M et al. doi: 10.3389/fnut.2022.881177. eCollection 2022., Martin LTP et al. doi: 10.1007/s11010-018-3464-z. Epub 2018 Oct 22., Rom DI et al. doi: 10.4193/Rhin16.181.,etc. We therefore revised as follows;

This review therefore focuses on the interaction of bitter and astringent tastes of polyphenols with sensory receptors, particularly with those that are expressed “extra-oral cavity”, which is the digestive tract other than the oral cavity.

Comment 5

Regarding the subtitles “3. Astringent Taste Receptors and Polyphenols  3.1. Mechanisms of Astringent Taste Perception” astringency is not recognized as a taste, it is considered a trigeminal mediated chemosensation!

- Response

We revised as follows and also checked other part;

  1. Astringent taste receptors and polyphenols

3.1.        Mechanisms of recognition for astringent tastcye perception

Comment 6

Regarding the statement: “Polyphenols are chemically classified as flavonoids (Figure. 1a) including flavanols, 151 flavanones, flavones, flavonols, flavanols, isoflavones, anthocyanins), chalcones, tannins 152 (proanthocyanidins, hydrolysable tannins, fluorotannins, complex tannins) and phenols 153 [55, 80].” Polyphenols are not only flavonoids, chalcones, tannins and phenols! There are several other classes of polyphenols, besides these : phenolic acids, polyphenolic amides, stilbenes, lignans, etc. Please see for ref Tsao R. Chemistry and biochemistry of dietary polyphenols. Nutrients. 2010 Dec;2(12):1231-46

-Response

We believe that there are currently various theories regarding the classification of polyphenols. For example, the largest database, “Phenol Explorer”, classifies polyphenols into 6 classes and 31 sub-classes, but this does not include tannins, which are polymers. Since this point is not the point of this paper, we have revised and added the citation as follows;

            Polyphenols are chemically classified as flavonoids (Figure. 1a) including flavanols, flavanones, flavones, flavonols, flavanols, isoflavones, anthocyanins), chalcones, lignans, tannins (proanthocyanidins, hydrolysable tannins, fluorotannins, complex tannins) and other phenolic derivatives; Refer to previous comprehensive reviews for the classification of polyphenols[19,81,82]

  1. Tsao, R., Chemistry and biochemistry of dietary polyphenols. Nutrients 2010, 2, (12), 1231-46.

Comment 7

Regarding Figure 1, the authors should present the basic chemical structure of all the classes of polyphenols (not only flavonoids), or no structures at all.

-Response

The structure of flavonoids was left unchanged as discussed in section 5.2 on blood glucose and polyphenols, while others were removed as per the instructions of Reviewer 2.

Comment 8

Some statements lack references. e.g. “flavanols, a type of astringent polyphenol,”

-Response

We have added cited papers (95,99,101).

Comment 9 and 10

-I do not understand why the a subsection about polyphenols and stress response in the section entitled “Astringent Taste Receptors and Polyphenols”? This shows a lack of focus, and has nothing to do with the taste receptors and the title…

-Why talk about capsaicin, a pungent amide alkaloid in a paper about polyphenols, in a section about astringent polyphenols?

- The connection between the two following statement should be imporved, because it is not clear for the reader the idea of the authors that astringency as a stress acts through similar mechanism with those initiated by exercise.

-Response

We found that procyanidins and anthocyanins, which have an astringent taste, induce stress responses in mammals by activating TRP channels. These findings followed the following process. First, astringent polyphenols were found to exhibit stress responses (increased sympathetic activity and activation of the HPA axis characterized by hormesis(To clarify, we added Figure 2). And then astringent polyphenols were found to activate TRP channels in the same way as other stressors such as capsaicin, change of temperature, fasting and exercise. Therefore, we assigned the following topics:  3.1 Astringent taste recognition mechanism (renamed Mechanisms of recognition for astringency perception), 3.2. Astringent Polyphenols and Stress Response (renamed to Astringency perception is a stressor), and 3.3.  Astringent polyphenols and TRP channels. It was difficult to understand how the subsections were connected, so I added some statement in text in 3.1. and 5.4. Please refer the manuscript for details.

Comment 11

   There is redundant information. For instance, in subsection 3.3. “According to the research results to date, it seems that astringency is likely to be a 306 somatic sensation [94, 95].” The authors have already talked about this in a previous subsection.

-Response

We revised as follows;

As mentioned in the previous chapter, astringency is likely to be a somatic sensation as with other stressors

.

Comment 12

 “Biological Regulation through Bitter and Astringency of Polyphenols” should be “Biological Regulation through Bitterness and Astringency of Polyphenols

-Response

We revised and checked other part.

Comment 13

Abbreviations should be explained at their first appearance in the text. e.g. FMD

-Response

We explained FMD in L260 and checked other part.

Comment 14

  Reformulate “polyphenols…migrate” in Conclusion section.

-Response

We revised.

Reviewer 3 Report

Comments and Suggestions for Authors

Major: Suggested changes

1.     The Title of the review is too long and could be shortened

2.     The Introduction section is too long and does not orient the reader to the theme of the review. Lines 86 through 116 may be retained in the Introduction section.

3.     However, Lines 1 through 86 can be separated in the Heading “Receptors involved in sensory nutrition.

Minor: Suggested changes in the text:

Abstract

1.     Line 16.   …..nociceptors

Introduction

2.     Line 44. ……(TRP) channels

3.     Line 50. In recent years, it has been discovered that chemosensory receptors are expressed outside of the oral nasal trigeminal system [14-16]. Please delete “responsible for flavor formation”

4.     Line 52. I suggest to start a new paragraph starting with “For instance…..

5.     Line 52……olfactory receptors are in sperm regulate…

6.     Lines 58-59. The bitter taste receptors, the taste receptors 2 (T2R) receptors, are expressed……

7.     Lines 66-67. Salt taste channels, the epithelial sodium channels (ENaCs), is known to be expressed in……..

Bitter Taste Receptors and Polyphenols

8.     Lines 122-123……seven transmembrane receptors expressed in taste receptor cells within the taste buds.

9.     Lines 151-152.  Flavanols is repeated twice.

10. Line 161. ….was conducted using the following method.

11. Line 180. Please define EGCG

12. Lines 208-209. Not clear what the authors means “Subsequent results could be applied to the discovery of bitter-taste medicines”. Recommend to delete this sentence.

13. Line 220. …taste receptors expressed in taste receptor cells in the taste buds.

14. Line 248…….GPCRs [94] or TRP channels [99-101].

Astringent Polyphenols and Stress Response

15.  Line 313.  …..(TRPV1) channel, which is…..

16. Line 346…..incubated….

Bioavailability of Polyphenols

17. Line 395. …excreted from the body in feces,….

Biological Regulation through Bitter and Astringency of Polyphenols

18. Line 406……controlling obesity…..

19. Line 423…..therefore, the main compounds cannot be limited (This sentence is not clear. Please modify).

Conclusions

20. Line 572. ….polyphenols is beneficial……

Comments on the Quality of English Language

Quality of English is good. However, I have suggested minor changes in the text.

Author Response

Thank you for conveying comments and reviewers’s suggestions. We appreciated the suggestions and revised the manuscript accordingly.

RESPONSE TO REVIEWER 3 :We wish to express our appreciation to Reviewer 1

Comment1

The Title of the review is too long and could be shortened

-Response

We changed titile to “ Sensory nutrition and bitterness and astringency of polyphenols”

Comment2

The Introduction section is too long and does not orient the reader to the theme of the review. Lines 86 through 116 may be retained in the Introduction section.

However, Lines 1 through 86 can be separated in the Heading “Receptors involved in sensory nutrition.

-Response

Minor: Suggested changes in the text:

Abstract

  1. Line 16.   …..nociceptors  → revised

Introduction

  1. Line 44. ……(TRP) channels → revised
  2. Line 50. In recent years, it has been discovered that chemosensory receptors are expressed outside of the oral nasal trigeminal system [14-16]. Please delete “responsible for flavor formation”  → revised
  3. Line 52. I suggest to start a new paragraph starting with “For instance…..
  4. Line 52……olfactory receptors arein sperm regulate…→ revised
  5. Lines 58-59. The bitter taste receptors, the taste receptors 2 (T2R) receptors, are expressed……→ revised
  6. Lines 66-67. Salt taste channels, the epithelial sodium channels(ENaCs), is known to be expressed in……..→ revised

Bitter Taste Receptors and Polyphenols

  1. Lines 122-123……seven transmembrane receptors expressed in taste receptor cells within the taste buds. → revised
  2. Lines 151-152.  Flavanols is repeated twice. → revised
  3. Line 161. ….was conducted usingthe following method. → revised
  4. Line 180. Please define EGCG → We stated it in L176.
  5. Lines 208-209. Not clear what the authors means “Subsequent results could be applied to the discovery of bitter-taste medicines”. Recommend to delete this sentence. → deleted
  6. Line 220. …taste receptors expressed in taste receptor cells in the taste buds. → we revised as “taste receptors expressed in taste cells in the taste buds.”
  7. Line 248…….GPCRs [94] or TRP channels [99-101]. → revised

Astringent Polyphenols and Stress Response

  1. Line 313.  …..(TRPV1) channel, which is…..→ revised
  2. Line 346…..incubated….→ revised

Bioavailability of Polyphenols

  1. Line 395. …excreted from the body infeces,…. → revised

Biological Regulation through Bitter and Astringency of Polyphenols

  1. Line 406……controlling obesity…..→ revised
  2. Line 423…..therefore, the main compounds cannot be limited (This sentence is not clear. Please modify). → we revised to “This is because it is difficult to identify the main compounds responsible for biological activity because many experiments have involved mixtures”.

Conclusions → revised

  1. Line 572. ….polyphenols isbeneficial……→ revised

Round 2

Reviewer 2 Report

Comments and Suggestions for Authors

Regarding some of my comments, I found no improvement. English errors may lead to scientific errors.

1) lines 136-138. "For example, in the gastrointestinal tract, by the activation of GPCRs, olfactory receptors and sweet/umami/bitter taste receptors, secrete gastrointestinal hormones from intestinal secretory cells" The subject of the statement should be secretory cells, not the receptors. The idea is that the receptors do not secrete, but the cells are the one that secrete.  Or "..activation of ....receptors leads to the secretion of gastrointestinal hormones by the intestinal secretory cells "

2) This subtitle needs to be reformulated: "2.1. Bitter taste receptors expressed extra-oral cavity and GI hormones."

3) "Astringent receptors and polyphenols" may be better "Astringency sensors and polyphenols"?

4) regarding my prebious comment 4 on the subtitle "Bitter Taste Receptors Expressed Extra-Oral Cavity and GI Hormones”, I must say that the authors are wrong when they state "The term "Extra-oral cavity" is frequently used in recent papers regarding taste receptors expressed in gut other than the oral cavity. for example, Ziegler F, et al. doi: 10.1038/s42003-023-04971-3, Behrens M et al. doi: 10.3389/fnut.2022.881177. eCollection 2022., Martin LTP et al. doi: 10.1007/s11010-018-3464-z. Epub 2018 Oct 22., Rom DI et al. doi: 10.4193/Rhin16.181.,etc." In these papers the term "extra-oral" is indeed frequently used in expression like "extra-oral tissue", "extra-oral taste receptors", "extra-oral disribution", but not "extra-oral cavity". Please, replace the "extra-oral cavity" with something else. A potential alternative may be "Extra-oral Bitter Taste Receptors and GI Hormones"

Author Response

RESPONSE TO REVIEWER 2

 Thank you for conveying comments and reviewer 2’s suggestions. We appreciated the suggestions and revised the manuscript accordingly.

Comment 1

lines 136-138. "For example, in the gastrointestinal tract, by the activation of GPCRs, olfactory receptors and sweet/umami/bitter taste receptors, secrete gastrointestinal hormones from intestinal secretory cells" The subject of the statement should be secretory cells, not the receptors. The idea is that the receptors do not secrete, but the cells are the one that secrete.  Or "..activation of ....receptors leads to the secretion of gastrointestinal hormones by the intestinal secretory cells "

- Response

We revised as follows.

From

For example, in the gastrointestinal tract, by the activation of GPCRs, sweet/umami/bitter taste receptors, secrete gastrointestinal hormones from intestinal secretory cells.

To

For example, in the gastrointestinal tract, by the activation of GPCRs,  sweet/umami/bitter taste receptors lead to the secretion of gastrointestinal hormones by the intestinal secretory cells.

Comment 2

This subtitle needs to be reformulated: "2.1. Bitter taste receptors expressed extra-oral cavity and GI hormones."

- Response

We revised as follows.

From

.Bitter taste receptors expressed extra-oral cavity and GI hormones

To

 Extra-oral bitter taste receptor and gastrointestinal hormones

Comment 3

"Astringent receptors and polyphenols" may be better "Astringency sensors and polyphenols"?

- Response

From

Astringent receptors and polyphenols

To

Astringency sensors and polyphenols

Comment 4

regarding my prebious comment 4 on the subtitle "Bitter Taste Receptors Expressed Extra-Oral Cavity and GI Hormones”, I must say that the authors are wrong when they state "The term "Extra-oral cavity" is frequently used in recent papers regarding taste receptors expressed in gut other than the oral cavity. for example, Ziegler F, et al. doi: 10.1038/s42003-023-04971-3, Behrens M et al. doi: 10.3389/fnut.2022.881177. eCollection 2022., Martin LTP et al. doi: 10.1007/s11010-018-3464-z. Epub 2018 Oct 22., Rom DI et al. doi: 10.4193/Rhin16.181.,etc." In these papers the term "extra-oral" is indeed frequently used in expression like "extra-oral tissue", "extra-oral taste receptors", "extra-oral disribution", but not "extra-oral cavity". Please, replace the "extra-oral cavity" with something else. A potential alternative may be "Extra-oral Bitter Taste Receptors and GI Hormones"

- Response

We revised as follows.

From

particularly with those that are expressed “extra-oral cavity”, which is the digestive tract other than the oral cavity.

To

particularly with those that are expressed in the digestive tract other than the oral cavity.